# Pretreatment Tumor Growth Rate and Radiological Response as Predictive Markers of Pathological Response and Survival in Patients with Resectable Lung Cancer Treated by Neoadjuvant Treatment

**DOI:** 10.3390/cancers15164158

**Published:** 2023-08-17

**Authors:** Toulsie Ramtohul, Léa Challier, Vincent Servois, Nicolas Girard

**Affiliations:** 1Department of Radiology, Institut Curie Paris, PSL Research University, 75005 Paris, France; lea.challier@curie.fr (L.C.); vincent.servois@curie.fr (V.S.); 2Institut du Thorax Curie Montsouris, Institut Curie, 75005 Paris, France; nicolas.girard2@curie.fr; 3Paris Saclay Campus, Versailles Saint Quentin University, 78000 Versailles, France

**Keywords:** non-small cell lung cancer, pretreatment tumor growth rate, event-free survival, overall survival, major pathological response

## Abstract

**Simple Summary:**

For patients with resectable non-small cell lung carcinoma (NSCLC), neoadjuvant nivolumab and chemotherapy are associated with increased major pathological responses and better event-free survival. Identification of earlier biomarkers associated with progression precluding surgery or disease recurrence after surgery is of importance in this population. The aim of our retrospective study was to assess the potential added value of pretreatment tumor growth rate (TGR_0_) using computed tomography (CT) and/or positron emission tomography (PET)-CT scans before and at baseline. We confirmed in 32 patients with resectable stage IB (≥4 cm) to IIIA NSCLC that the assessment of TGR_0_ helps identify patients who would benefit from neoadjuvant treatment and outperforms RECIST assessments for survival outcomes. TGR_0_ may be an early noninvasive marker for more favorable genetic and/or biological profiles, leading to improved disease control and overall survival.

**Abstract:**

Introduction: Predictive biomarkers associated with pathological response, progression precluding surgery, and/or recurrence after surgery are needed for patients with resectable non-small cell lung carcinoma (NSCLC) treated by neoadjuvant treatment. We evaluated the clinical impact of the pretreatment tumor growth rate (TGR_0_) and radiological response for patients with resectable NSCLC treated with neoadjuvant therapies. Methods: Consecutive patients with resectable stage IB (≥4 cm) to IIIA NSCLC treated by neoadjuvant platinum-doublet chemotherapy with or without nivolumab at our tertiary center were retrospectively analyzed. TGR_0_ and RECIST objective responses were determined. Multivariable analyses identified independent predictors of event-free survival (EFS), overall survival (OS), and major pathological response (MPR). Results: Between November 2017 and December 2022, 32 patients (mean [SD] age, 63.8 [8.0] years) were included. At a median follow-up of 54.8 months (95% CI, 42.3–60.4 months), eleven patients (34%) experienced progression or recurrence, and twelve deaths (38%) were recorded. The TGR_0_ cutoff of 30%/month remained the only independent factor associated with EFS (HR = 0.04; 95% CI, 0.01–0.3; *p* = 0.003) and OS (HR = 0.2; 95% CI, 0.03–0.7; *p* = 0.01). The TGR_0_ cut-off had a mean time-dependent AUC of 0.83 (95% CI, 0.64–0.95) and 0.80 (95% CI, 0.62–0.97) for predicting EFS and OS, respectively. Fifteen of 26 resection cases (58%) showed MPR including nine with pathological complete responses (35%). Only the objective response of the primary tumor was associated with MPR (OR = 27.5; 95% CI, 2.6–289.1; *p* = 0.006). Conclusions: Assessment of TGR_0_ can identify patients who should benefit from neoadjuvant treatment. A tumor objective response might be a predictor of MPR after neoadjuvant treatment, which will help to adapt surgical management.

## 1. Introduction

Lung cancer is the leading cause of cancer-related death worldwide [1]. Due to the increased use of screening CT scans in high-risk patients, the proportion of non-small cell lung cancer (NSCLC) diagnosed at an early-stage increases up to 30% [2,3,4]. Surgery remains the best treatment modality for curing patients diagnosed with resectable stage I-IIIA NSCLC [5]. However, more than 50% of patients with resectable NSCLC will experience recurrence after surgery alone [6,7,8]. Adjuvant chemotherapy is associated with an improved recurrence-free survival for stage IB-IIIA patients, resulting in absolute survival benefits of 5.4% to 6.9% at five years [9,10]. Neoadjuvant chemotherapy provides few significant pathological responses and a comparable level of risk reduction compared to adjuvant chemotherapy [11]. Recently, the CheckMate 816 trial, which evaluated the combination of neoadjuvant nivolumab and chemotherapy in 358 newly diagnosed patients with resectable stage IB to IIIA NSCLC, reported increased major and complete pathological responses and better event-free survival (EFS) compared to chemotherapy alone, without more adverse effects on surgical feasibility or surgical outcomes [12]. However, 15% to 20% of patients do not undergo definitive surgery, and predictive factors of the long-term benefits of neoadjuvant treatment are still under investigation [13]. Overall, predicting pathological complete response (pCR) and major pathological responses (MPR) is a challenge for optimizing surgical approaches.

The tumor growth rate (TGR) provides a means of quantitative evaluation of tumor volume changes over time that may be calculated before treatment onset, leading to a better understanding of natural growth kinetics [14]. TGR-derived parameters have been validated as radiological markers of progression-free survival and overall survival in different cancer types [15,16]. Pretreatment TGR is associated with inferior progression-free survival and distant control among patients with locally advanced NSCLC and helps in identifying hyperprogressive disease in patients treated with PD-1/PD-L1 [17,18]. The aim was to evaluate the clinical impact of pretreatment TGR on the survival outcomes and pathological responses of patients with resectable NSCLC treated with neoadjuvant chemotherapy with or without nivolumab.

## 2. Materials and Methods

### 2.1. Patients and Study Design

This single-center study was approved by the institutional ethics review boards, and written informed consent was obtained from all patients. Between November 2017 and December 2022, consecutive patients with resectable NSCLC treated by neoadjuvant treatment at our tertiary center were retrospectively analyzed. Key eligibility criteria included age older than or equal to 18 years, histologically confirmed resectable stage IB (≥4 cm) to IIIA NSCLC (according to the staging criteria of the American Joint Committee on Cancer, 7th edition), Eastern Cooperative Oncology Group performance status 0 to 1, no previous anticancer therapy or measurable disease per Response Evaluation Criteria in Solid Tumors (RECIST) version 1.1. Patients with known ALK translocations or EGFR mutations were excluded. Patients received nivolumab (360 mg) plus platinum-doublet chemotherapy or platinum-doublet chemotherapy alone (every 3 weeks for three cycles) before undergoing definitive surgery. This study complied with the tenets of the Declaration of Helsinki.

### 2.2. Radiological Assessment

Tumors were assessed using computed tomography (CT) and/or positron emission tomography (PET)-CT scans before baseline, at baseline, and within 14 days prior to definitive surgery per response evaluation criteria in solid tumors (RECIST) v1.1. Examinations were centrally reviewed by two senior radiologists blinded to the clinical characteristics, treatment received, and outcomes. TGR_0_ is expressed as the percentage change in tumor volume over 1 month (%/m): TGR_0_ = 100 × [exp(TG) − 1], where TG = 3 × log(D2/D1)/time (months). D1 and D2 represent the largest diameter of the primary tumor according to RECIST v1.1 on pretreatment and baseline imaging, respectively. Lymph nodes were not taken into account in the TGR_0_ calculation. Detailed examples of TGR calculations are given in Figure 1. For each patient, the same imaging technique (CT/PET-CT scans) was preferred for each time point. The RECIST objective response was defined as the proportion of patients who experienced a complete response (CR) or partial response (PR) of the primary tumor after neoadjuvant treatment.

### 2.3. Survival Endpoints

Event-free survival (EFS) was calculated as the time from neoadjuvant treatment start to the occurrence of any radiologically identified disease progression precluding surgery, disease recurrence after surgery, disease progression in the absence of surgery, or death from any cause. Overall survival (OS) was calculated as the time from neoadjuvant treatment start to death from any cause.

### 2.4. Pathological Endpoints

An expert pathologist, unaware of patient characteristics and outcomes, reviewed hematoxylin and eosin-stained slides containing sections of the gross residual tumor post-surgery. The assessment involved comparing the estimated cross-sectional area of viable tumor with that of necrosis, inflammation, and fibrosis on each slide to determine the percentage of residual tumor. A major pathological response (MPR) was characterized as having a residual viable tumor of ≤10% within the primary tumor and sampled lymph nodes. Meanwhile, a complete pathological response (pCR) was identified when no residual viable tumor was present in either the primary tumor or lymph node tissue.

### 2.5. Statistical Analysis

Continuous variables were assessed utilizing the Wilcoxon–Mann–Whitney and Student tests, taking into account the normality of their distribution. Categorical variables underwent assessment through either the χ^2^ test or Fisher’s exact test. The inter-reader reliability of TGR_0_ was assessed with the intraclass correlation coefficient (ICC). To illustrate time-to-event outcomes, the Kaplan–Meier product-limit approach was utilized, and a comparison between the curves was determined using the exact log-rank test. For patients who either did not experience an event or were alive on the specified date (2 August 2023), their data were censored at the most recent evaluation date. The identification of the optimal TGR_0_ cutoff, differentiating patients based on EFS and OS, was carried out using a stepwise log-rank test. A multivariable Cox analysis was conducted, initially including all variables associated with EFS or OS from the univariate analysis at a significance level of *p* < 0.05. The assumption of proportional hazards was verified using Schoenfeld residuals. All computations were conducted using SAS software (version 9.4, SAS Institute, Cary, NC, USA). Statistical tests were two-tailed, and *p*-values below 0.05 were considered indicative of statistical significance.

## 3. Results

### 3.1. Patient Characteristics

Between November 2017 and December 2022, 32 patients (mean [SD] age, 63.8 [8.0] years) were identified in our database. All patients had at least one available imaging scan before baseline. The baseline characteristics are summarized in Table 1. Twenty-seven patients (84%) had stage IIIa disease. Neoadjuvant nivolumab-based therapy was delivered to 23 patients (72%). Twenty-six patients (81%) had low-TGR_0_ (≤30%/month) and six (19%) high-TGR_0_ (>30%/month). The median time between pretreatment and baseline imaging for the calculation of TGR_0_ was 1.9 months (IQR: 1.3–2.3). Patients with low-TGR_0_ had higher rates of non-squamous histology type (85% vs. 33%, *p* = 0.009) and disease stage IIIa (92% vs. 50%, *p* = 0.01) compared to those with high-TGR_0_. The reproducibility of the assessment for TGR_0_ among the two readers was very good with an ICC of 0.81 (95% CI: 0.76–0.95).

### 3.2. Prognostic Factors Associated with EFS and OS

The objective response rate of the primary tumor was 44% (14 of 32) (Figure 2). At the last database lock (2 August 2023), the median follow-up was 54.8 months (95% CI, 42.3–60.4 months). Overall, eleven patients (34%) experienced progression or recurrence, and twelve deaths (38%) were recorded. Seven patients (22%) had disease progression at the end of neoadjuvant treatment precluding surgery and four patients (13%) experienced lung cancer-related recurrence after surgery; two patients (6%) had a non-cancer death after surgery. The median EFS was not reached while the median OS was 54.1 months (95% CI, 23.7 months-not reached). There was a moderate difference in EFS (*p* = 0.045) but no significant difference in OS (*p* = 0.13) based on the RECIST objective response of the primary tumor during neoadjuvant treatment (Figure 3A,B). There was a larger difference in EFS (*p* < 0.001) and OS (*p* < 0.001) based on TGR_0_ before neoadjuvant treatment (Figure 3C,D). For patients receiving nivolumab-based neoadjuvant treatment, similar EFS and OS patterns were observed according to the TGR_0_ or objective response (Appendix A).

For EFS, a non-squamous histology type (HR = 0.3; 95% CI, 0.1–0.09; *p* = 0.047), a PD-L1 > 10% (HR = 0.2; 95% CI, 0.1–0.7; *p* = 0.01), a disease stage IIIa (HR = 0.3; 95% CI, 0.1–0.9; *p* = 0.04) and a low-TGR_0_ (HR = 0.04; 95% CI, 0.01–0.2; *p* < 0.001) were associated with higher EFS in univariable analyses and were included in the multivariable analysis (Table 2). Only the low-TGR_0_ remained an independent factor associated with higher EFS (HR = 0.02; 95% CI, 0.01–0.3; *p* = 0.003).

For OS, univariable analyses showed that a low-TGR_0_ (HR = 0.1; 95% CI, 0.02–0.4; *p* = 0.001) a PD-L1 > 10% (HR = 0.2; 95% CI, 0.1–0.8; *p* = 0.08) and disease stage IIIa (HR = 0.1; 95% CI, 0.02–0.4; *p* = 0.002) were associated with higher OS and were thus included in the multivariable analysis (Table 3). Only the low-TGR_0_ remained an independent factor associated with higher OS (HR = 0.2; 95% CI, 0.03–0.7; *p* = 0.01). The TGR_0_ cutoff had a mean time-dependent AUC of 0.83 (95% CI, 0.64–0.95) and 0.80 (95% CI, 0.62–0.97) for predicting EFS and OS, respectively (Figure 4).

### 3.3. Predictors of Pathological Response after Neoadjuvant Treatment

Twenty-six patients (81%) underwent surgery after neoadjuvant treatment and all resected patients had an R0 resection. Among them, 15 (58%) had MPR, including nine with pCR (36%). MPR was observed for both PD-L1–positive and PD-L1–negative tumors and was associated with both EFS (log-rank *p* = 0.002) and OS (log-rank *p* = 0.01) (Appendix A). Only an objective response of the primary tumor was associated with MPR (OR = 27.5; 95% CI, 2.6–289.1; *p* = 0.006) with a sensitivity and specificity of 0.73 (11/15 patients; [95% CI: 0.51, 0.96]) and 0.91 (10/11 patients; [95% CI: 0.74, 0.99]), respectively (Table 4 and Figure 5).

## 4. Discussion

Identification of earlier biomarkers associated with progression precluding surgery or disease recurrence after surgery is of importance in patients with resectable NSCLC treated by neoadjuvant treatment [19]. In our study, a lower pretreatment tumor growth rate (TGR_0_) was a strong factor associated with longer event-free and overall survivals after neoadjuvant treatment. The TGR_0_ could provide an early, noninvasive, cost-effective, and time-efficient method to identify patients likely to benefit from a neoadjuvant strategy.

Among patients with advanced lung cancer, the tumor growth rate has been recognized as an important marker of tumor response and progression, especially in the setting of immune checkpoint inhibitor therapy [17,18,20]. He et al. found that patients with metastatic NSCLC undergoing immunotherapy and exhibiting a high pretreatment TGR_0_ (> 25%/month) had lower PFS and a less durable clinical rate [20]. Interestingly, the threshold of TGR_0_ that best separated the groups with distinct clinical outcomes was comparable with the result in our study and consistent with a tumor volume doubling time of 80 days. In our study, all patients exhibiting a high pretreatment tumor growth rate (> 30%/month) experienced progression or recurrence of the disease and lower OS, indicating that neoadjuvant strategies including anti-PD-1 antibodies are not capable of inhibiting rapidly growing tumors, eradicating micrometastatic disease, and preventing tumor relapse [21,22,23].

The RECIST objective response of the primary tumor correlated moderately with clinical outcomes. Among patients without progression before surgery or recurrence after surgery, almost half did not show an objective response. The TGR_0_ model outperformed RECIST assessments with regard to survival prediction (time-dependent AUC of OS: 0.80 vs. 0.66) and helped to identify a large subset of patients (81%) with a low rate of progression or recurrence. In addition, we reported no significant correlation between smoking status, histologic type, disease stage or PD-L1 expression and survival outcomes. Our results are in accordance with the NADIM trial in which no strong association between the tumor response to treatment according to RECIST criteria or PD-L1 expression and survival was found [13]. In their trial, undetectable ctDNA at the end of neoadjuvant treatment has been proposed as a surrogate endpoint for long-term outcomes. However, the lack of standardization and the limited sensitivity of the current detection methods remained an obstacle for widespread clinical application [24]. TGR_0_ may be more biologically and clinically relevant for predicting patient clinical outcomes. Uncontrolled tumor growth is associated with a larger tumor burden, aberrant vascularization, and an altered immune microenvironment unfavorable for the action of PD-1 axis inhibitors [25,26].

A major pathologic response defined as ≤ 10% residual viable tumor in the primary tumor and lymph nodes was seen in almost 60% of patients receiving neoadjuvant treatment, mainly with nivolumab, which was consistent with other studies [12,13,27,28]. A major pathologic response might represent a promising surrogate endpoint for survival outcomes for patients with resectable tumors after neoadjuvant treatment [29,30,31]. A link between MPR and survival outcomes was also observed in this study. Additionally, we found that the RECIST objective response was the only variable associated with MPR. Indeed, neither clinical variables nor PD-L1 staining predicted MPR. This may have significant implications for optimizing the approach and extent of surgical resection as well as radiological follow-up after surgery or for adjuvant decisions [32].

Our study has several limitations. First, the study was retrospectively conducted at a single institute with a small sample size and a substantial number of exploratory variables; thus, results from multivariable analyses should be taken with caution. Second, a limitation of assessment of tumor growth dynamics is requirement of one imaging scan before baseline. In our experience, treatment decision-making often requires consecutive examinations separated by at least one month to confirm or complete staging information of NSCLC (e.g., lung CT and PET-CT scans). Additionally, frequent delays between initial imaging and screening often lead to the necessity of performing an additional staging scan immediately before neoadjuvant treatment. Third, the TGR assessment was based on the largest axis of the primary tumor, like the time-efficient approach of RECIST v1.1, although it may not reflect the whole tumor burden as non-targets like lymph nodes were not considered. In addition, automatic segmentation tools could be a more robust approach with less variability for volume growth assessment but were not investigated.

## 5. Conclusions

Pretreatment tumor growth rate (TGR_0_) provides information to select patients with slow-growing non-small cell lung cancer who should benefit from first-line neoadjuvant treatment. TGR_0_ may be an early radiographic marker for more favorable genetic and/or biologic profiles that result in improved disease control and overall survival. The objective response of the primary tumor has the potential to serve as a surrogate of a major pathological response.

## Figures and Tables

**Figure 1 cancers-15-04158-f001:**
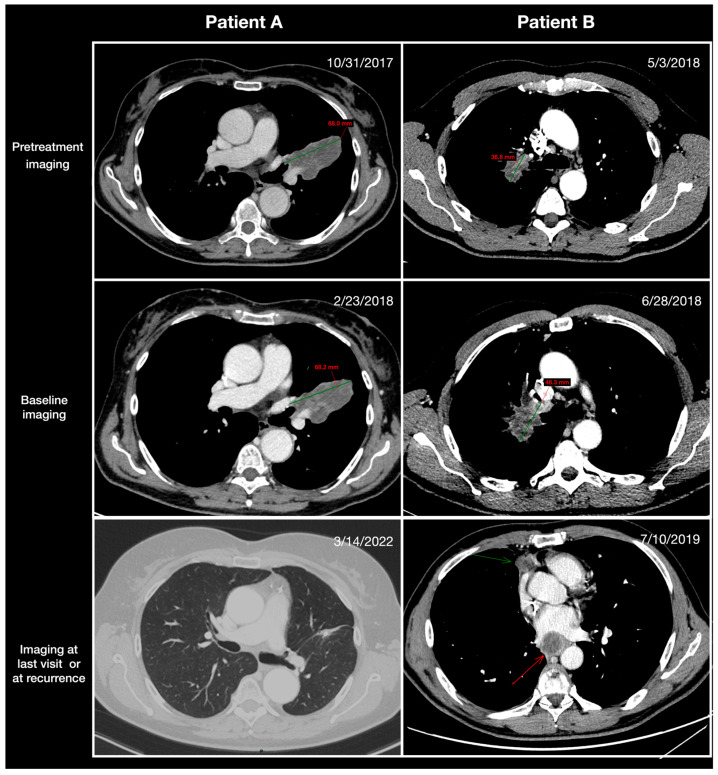
Illustration of the pretreatment tumor growth rate (TGR_0_) in two patients treated with neoadjuvant therapies for resectable non-small cell lung cancer (NSCLC). Patient A: Lung CT scan of a 61-year-old male patient who received neoadjuvant chemoimmunotherapy for non-squamous NSCLC. The tumor manifested as a proximal mass in the left upper lobe and was classified (according to the staging criteria of the American Joint Committee on Cancer, 7th edition) as stage IIIa NSCLC. Pretreatment imaging examinations revealed no significant growth rate of the lung mass with a TGR_0_ of 0.2%/month. The patient underwent a left upper lobectomy, with no evidence of recurrence after a follow-up of 46 months. Patient B: Lung CT scan of a 68-year-old male patient who received neoadjuvant chemoimmunotherapy for squamous NSCLC. The tumor manifested as a proximal mass in the right upper lobe and was classified (according to the staging criteria of the American Joint Committee on Cancer, 7th edition) as stage IIb NSCLC. Pretreatment imaging examinations revealed a significant growth rate of the lung mass with a TGR_0_ of 52%/month. The patient underwent a right upper lobectomy. Mediastinal necrotic lymph nodes (arrows) appeared 9 months after surgery, and the patient died 13 months after surgery.

**Figure 2 cancers-15-04158-f002:**
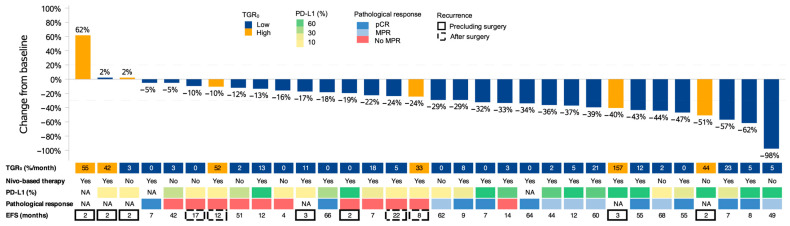
Change from baseline in tumor size, with patient data on pretreatment tumor growth rate (TGR_0_), baseline PD-L1 status, type of neoadjuvant treatment, pathological response, and time to event-free survival (months). The change from baseline (%) is labeled on each bar. Low-TGR_0_: ≤30%/month; high-TGR_0_: >30%/month; NA: not available.

**Figure 3 cancers-15-04158-f003:**
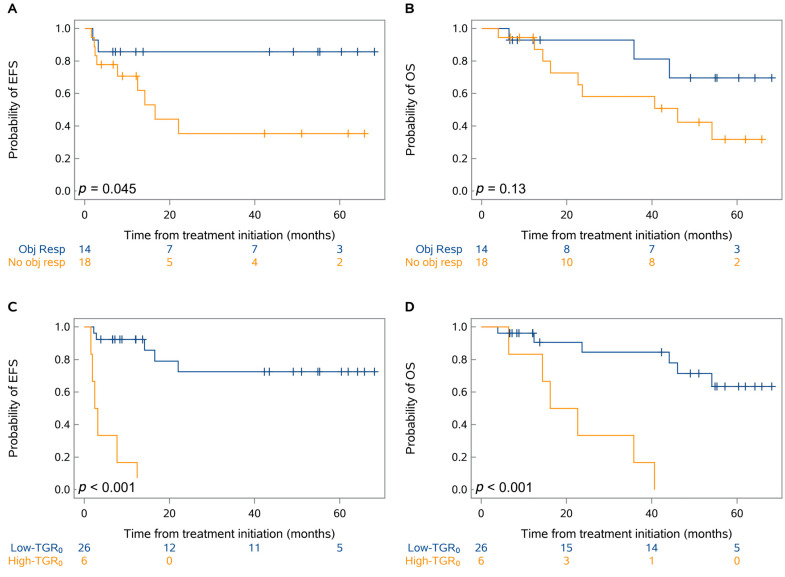
Kaplan-Meier analysis of event-free survival (EFS) and overall survival (OS) by RECIST objective response of the primary tumor (**A**,**B**) and pretreatment TGR_0_ (**C**,**D**). *p*-values were obtained using the log-rank test.

**Figure 4 cancers-15-04158-f004:**
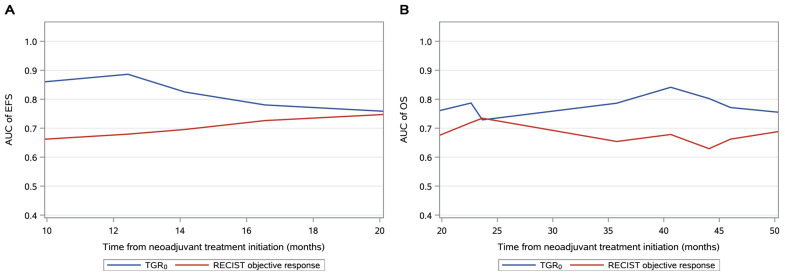
A time-dependent area under the TGR_0_ curve compared with the RECIST objective response of the primary tumor for event-free survival (**A**) and overall survival (**B**). The TGR_0_ model had a mean time-dependent AUC of 0.83 (95% CI, 0.64–0.95) and 0.80 (95% CI, 0.62–0.97) for predicting EFS and OS, respectively. The RECIST objective response model had a mean time-dependent AUC of 0.70 (95% CI, 0.46–0.94) and 0.66 (95% CI, 0.43–0.88) for predicting EFS and OS, respectively.

**Figure 5 cancers-15-04158-f005:**
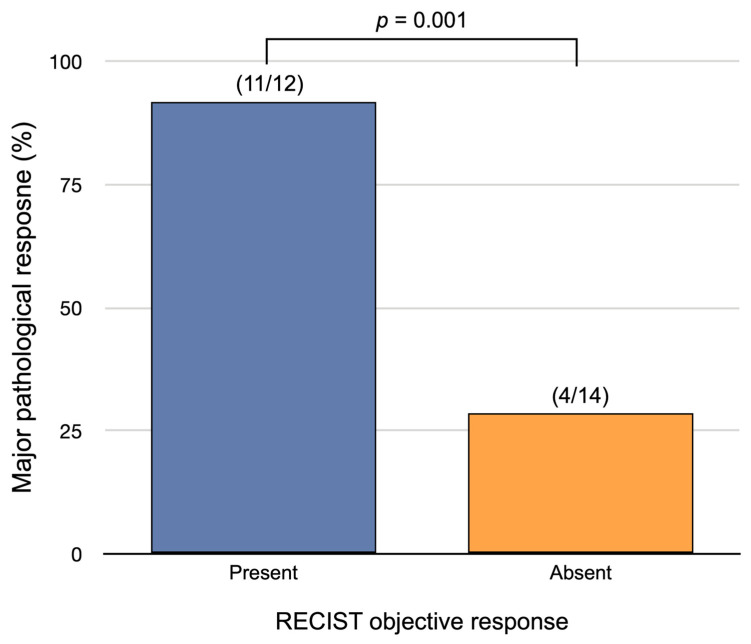
Correlation of major pathological response and RECIST objective response of the primary tumor.

**Table 1 cancers-15-04158-t001:** Baseline characteristics. Data are numbers of patients with percentages in parentheses. Disease stage was based on TNM Classification of Malignant Tumors, 7th edition. Objective response was defined as the proportion of patients with a complete response or partial response of the primary tumor according to RECIST v1.1. A major pathological response was available for 26 patients. * Two patients with low-TGR_0_ had missing data for PD-L1 status. *P*-values were obtained using the χ^2^ test. Low-TGR_0_: ≤30%/month; high-TGR_0_: >30%/month.

Variables		Whole Cohort	Pretreatment TGR_0_		*p*-Value
		(*n* = 32)	Low (*n* = 26)	High (*n* = 6)	
Age (years)	≤60	11 (34%)	7 (27%)	4 (67%)	0.07
	>60	21 (66%)	19 (73%)	2 (33%)	
Sex	Female	10 (31%)	8 (31%)	2 (33%)	0.90
	Male	22 (69%)	18 (69%)	4 (67%)	
ECOG performance status	0	27 (84%)	22 (85%)	5 (83%)	0.94
	1	5 (16%)	4 (15%)	1 (17%)	
Smoking status	Current or former smoker	29 (91%)	24 (92%)	5 (83%)	0.50
	Never smoked	3 (9%)	2 (8%)	1 (17%)	
Histologic type	Non-squamous	24 (75%)	22 (85%)	2 (33%)	0.009
	Squamous	8 (25%)	4 (15%)	4 (67%)	
PD-L1 status (%) *	≤10	11 (37%)	7 (29%)	4 (67%)	0.09
	>10	19 (63%)	17 (71%)	2 (33%)	
Disease stage	Ib or II	5 (16%)	2 (8%)	3 (50%)	0.01
	IIIa	27 (84%)	24 (92%)	3 (50%)	
Largest tumor size at baseline (mm)	≤50	19 (59%)	16 (62%)	3 (50%)	0.60
	>50	13 (41%)	10 (38%)	3 (50%)	
Nodal stage	N1/2	28 (88%)	23 (88%)	5 (83%)	0.73
	N0	4 (13%)	3 (12%)	1 (17%)	
Nivolumab-based neoadjuvant treatment	Present	23 (72%)	18 (69%)	5 (83%)	0.49
	Absent	9 (28%)	8 (31%)	1 (17%)	
RECIST objective response	Present	14 (44%)	12 (46%)	2 (33%)	0.57
	Absent	18 (56%)	14 (54%)	4 (67%)	
Major pathological response	Present	15 (58%)	15 (63%)	0 (0%)	0.09
	Absent	11 (42%)	9 (38%)	2 (100%)	

**Table 2 cancers-15-04158-t002:** Multivariable analysis of EFS. Multivariable analysis was undertaken by entering all variables at the *p* < 0.05 level in the univariable analysis. HR: hazard ratio; EFS: event-free survival; CI: confidence interval.

Variable	Univariable Analysis of EFS		Multivariable Analysis of EFS	
	HR (95% CI)	*p*-Value	HR (95% CI)	*p*-Value
Age (years), >60 vs. ≤60	0.6 (0.2–1.8)	0.33		
Sex, female vs. male	0.8 (0.2–3.2)	0.80		
Smoking status, never smoked vs. current or former smoker	4.7 (0.9–25.9)	0.07		
Histologic type, non-squamous vs. squamous	0.3 (0.1–0.9)	0.047	0.6 (0.1–4.2)	0.58
PD-L1 (%), >10 vs. ≤10	0.2 (0.1–0.7)	0.01	0.3 (0.1–1.4)	0.13
Disease stage, IIIa vs. Ib/II	0.3 (0.1–0.9)	0.04	5.2 (0.5–58.9)	0.18
Largest tumor size at baseline (mm), >50 vs. ≤50	0.8 (0.2–2.9)	0.78		
Nodal stage, N1/2 vs. N0	1.2 (0.1–9.3)	0.88		
Nivolumab-based treatment, present vs. absent	1.3 (0.3–4.9)	0.72		
RECIST objective response, present vs. absent	0.2 (0.1–1.1)	0.07		
TGR_0_ (%/month), ≤30 vs. >30	0.04 (0.01–0.2)	<0.001	0.04 (0.01–0.3)	0.003

**Table 3 cancers-15-04158-t003:** Multivariable analysis of OS. Multivariable analysis was undertaken by entering all variables at the *p* < 0.05 level in the univariable analysis. HR: hazard ratio; OS: overall survival; CI: confidence interval.

Variable	Univariable Analysis of OS		Multivariable Analysis of OS	
	HR (95% CI)	*p*-Value	HR (95% CI)	*p*-Value
Age (years), >60 vs. ≤60	0.9 (0.3–3.1)	0.90		
Sex, female vs. male	1.2 (0.4–4.0)	0.76		
Smoking status, never smoked vs. current or former smoker	-	>0.99		
Histologic type, non-squamous vs. squamous	0.4 (0.1–1.2)	0.10		
PD-L1 (%), >10 vs. ≤10	0.2 (0.1–0.8)	0.02	0.4 (0.1–1.6)	0.19
Disease stage, IIIa vs. Ib/II	0.2 (0.1–0.7)	0.02	0.5 (0.1–1.9)	0.29
Largest tumor size at baseline (mm), >50 vs. ≤50	1.0 (0.3–3.1)	0.97		
Nodal stage, N1/2 vs. N0	0.8 (0.1–6.2)	0.82		
Nivolumab-based treatment, present vs. absent	1.2 (0.4–4.1)	0.76		
RECIST objective response, present vs. absent	0.4 (0.1–1.4)	0.15		
TGR_0_ (%/month), ≤30 vs. >30	0.1 (0.02–0.4)	0.001	0.2 (0.03–0.7)	0.01

**Table 4 cancers-15-04158-t004:** Multivariable analysis of major pathological response. Multivariable analysis was undertaken by entering all variables at the *p* < 0.05 level in the univariable analysis. MPR: major pathological response; OR: odds ratio; CI: confidence interval.

Variable	Univariable Analysis of MPR		Multivariable Analysis of MPR	
	OR (95% CI)	*p*-Value	OR (95% CI)	*p*-Value
Age (years), >60 vs. ≤60	1.1 (0.2–5.8)	0.87		
Sex, female vs. Male	0.4 (0.1–2.3)	0.32		
Smoking status, never smoked vs. current or former smoker	-	>0.99		
Histologic type, non-squamous vs. squamous	1.5 (0.2, 9.4)	0.66		
PD-L1 (%), >10 vs. ≤10	3.1 (0.6–17.3)	0.20		
Disease stage, IIIa vs. Ib/II	5.2 (0.5–59.3)	0.18		
Largest tumor size at baseline (mm), >50 vs. ≤50	3.9 (0.6–24.7)	0.14		
Nodal stage, N1/2 vs. N0	0.7 (0.1–8.2)	0.74		
Nivolumab-based treatment, present vs. absent	2.3 (0.4–13.3)	0.36		
RECIST objective response, present vs. absent	27.5 (2.6–289.1)	0.006	27.5 (2.6–289.1)	0.006
TGR_0_ (%/month), ≤30 vs. >30	-	>0.99		

## Data Availability

The data presented in this study are available on request from the corresponding author.

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
