# Peer review of "Pretreatment Tumor Growth Rate and Radiological Response as Predictive Markers of Pathological Response and Survival in Patients with Resectable Lung Cancer Treated by Neoadjuvant Treatment"

_cancers, 2023, doi:10.3390/cancers15164158_

Round 1

Reviewer 1 Report

GENERAL COMMENTS

1.       The idea of determining the pre-treatment tumor growth rate has merit, but the limitations of this study are important.

2.       The study is well written, but the amount of original results is limited mainly because of the small sample size, and the recruitment from a single centre.

3.       Page 3: “Between November 2017 and April 2022, 25 patients (mean [SD] age, 64.2 [7.8] years) were identified in our database. [……………………..]  Overall, nine patients (36%) experienced progression or recurrence, and eight deaths (32%) were recorded.”   My main reason of concern with this paper refers to these two sentences: a total of 25 patients were studied; among these patients, eight or nine events (deaths or progressions, respectively) were recorded.  While the objective of this paper is to propose recommendations about  predictive markers of pathological response and survival, the question arises of how these markers can be based on a very small number of events. It is likely that changing the number of events by one event more or one event less would determine profound consequences on the results.

4.       In my view, the finding that some differences (e.g. favorable vs intermediate vs unfavorable) were statistically significant reflects a difference question compared with the predictive power of these three group assignments. In practice, the rate of false positive and  false negative assignments could be more important or, in technical terms, determining specificity and sensitivity could be worthwhile.

5.       On the one hand, the groups are three (favorable, intermediate, unfavorable), but on the other hand the HRs are binary (i.e.: HR of favorable vs intermediate; HR of favorable vs unfavorable; HR of intermediate vs unfavorable). This complicates the interpretation of these data. Using the medians as opposed to the HRs would make these interpretations much simpler.

6.       Page 5: “Our study has several limitations. First, the study was retrospectively conducted at a single institute with a small sample size.”
I agree.

SPECIFIC COMMENTS

7.       None.

None

Reviewer 2 Report

1. Figure 5 isn't necessary for the date, if you need to add some date like MPR, I advise you to link the pathologic response(like MPR rate) with TGR0

2.3/5 cases for the unfavorable group without the date of PDL1 expression rate and pathological response date. you need to complete this 

3、need to add more cases, especially your unfavorable group

  •  

Round 2

Reviewer 1 Report

GENERAL COMMENTS

1.      After defining the pretreatment tumor growth rate (TGR0) has been defined as  an index based computed tomography (CT) and/or positron emission tomography (PET)-CT scans  before and at baseline, the authors evaluate whether this index is helpful to identify patients who benefit from neoadjuvant treatment. The proposed cutoff at 30 percent/month is shown to correlate with EFS. Six patients had a value above this cutoff.

2.      This study is based on only 32 patients; neoadjuvant nivolumab-based therapy was delivered to 23 patients (72%). Six patients had a value of the proposed index above the cutoff of 30 percent/month.

3.      In the Discussion, more emphasis should be placed on the limitation represented by the small number of included patients.

4.      Overall, the study is methodologically correct, and also reports an interesting finding. Its main limitation is represented by the very small number of patients included in the analysis.

5.      The decision on the acceptance of this paper depends more on the relevance of its findings than on the quality of its methods, which is adequate.

SPECIFIC COMMENTS

1.      Page 1, Line 18: “help” should be “helps”

6.      Page 5, lines 193-195: Regarding the sentence “Among patients with advanced lung cancer, the tumor growth rate has been recognized as an important marker of tumor response and progression, especially in the setting of immune checkpoint inhibitor therapy”, please add all the pertinent references (i.e. 20-24 or at least tose most pertinent among these 5 references) at the end of the sentence.

2.      Page 5, line 200: add percent after 30.

Reviewer 2 Report

Your diagram has good improvement which could prove your TGR0
